# Serum miRNA Profiling for Early PDAC Diagnosis and Prognosis: A Retrospective Study

**DOI:** 10.3390/biomedicines9070845

**Published:** 2021-07-20

**Authors:** Ada Aita, Caterina Millino, Cosimo Sperti, Beniamina Pacchioni, Mario Plebani, Cristiano De Pittà, Daniela Basso

**Affiliations:** 1Department of Medicine-DIMED, University of Padova, 35128 Padova, Italy; ada.aita@unipd.it (A.A.); mario.plebani@unipd.it (M.P.); 2Department of Biology, University of Padova, 35131 Padova, Italy; caterina.millino@unipd.it (C.M.); beniamina.pacchioni@unipd.it (B.P.); 3Department of Surgical, Oncological and Gastroenterological Sciences-DiSCOG, University of Padova, 35128 Padova, Italy; cosimo.sperti@unipd.it

**Keywords:** PDAC, miRNAs, survival, expression profiling

## Abstract

Background: Tumor stage predicts pancreatic cancer (PDAC) prognosis, but prolonged and short survivals have been described in patients with early-stage tumors. Circulating microRNA (miRNA) are an emerging class of suitable biomarkers for PDAC prognosis. Our aim was to identify whether serum miRNA signatures predict survival of early-stage PDAC. Methods: Serum RNA from archival 15 stage I-III PDAC patients and 4 controls was used for miRNAs expression profile (Agilent microarrays). PDAC patients with comparable age, gender, diabetes, jaundice and surgery were classified according to survival: less than 14 months (7/15 pts, group A) and more than 22 months (8/15 pts, group B). Bioinformatic data analysis was performed by two-class Significance Analysis of Microarray (SAM) algorithm. Binary logistic regression analyses considering PDAC diagnosis and outcome as dependent variables, and ROC analyses were also performed. Results: 2549 human miRNAs were screened out. At SAM, 76 differentially expressed miRNAs were found among controls and PDAC (FDR = 0.4%), the large majority (50/76, 66%) of them being downregulated in PDAC with respect to controls. Six miRNAs were independently correlated with early PDAC, and among these, *hsa-miR-6821-5p* was associated with the best ROC curve area in distinguishing controls from early PDAC. Among the 71 miRNAs differentially expressed between groups A and B, the most significant were *hsa-miR-3135b* expressed in group A only, *hsa-miR-6126* and *hsa-miR-486-5p* expressed in group B only. Eight miRNAs were correlated with the presence of lymph-node metastases; among these, *hsa-miR-4669* is of potential interest. *hsa-miR-4516*, increased in PDAC and found as an independent predictor of survival, has among its putative targets a series of gens involved in key pathways of cancer progression and dissemination, such as Wnt and p53 signalling pathways. Conclusions: A series of serum miRNAs was identified as potentially useful for the early diagnosis of PDAC, and for establishing a prognosis.

## 1. Introduction

The incidence of pancreatic ductal adenocarcinoma (PDAC) is increasing worldwide, this disease being among the first 15 leading forms of cancer in men and women, with its highest incidence rates in Europe, Northern America, and Australia/New Zealand [1,2].

Despite advances made in the last few decades in understanding PDAC biology and developing new targeted therapies [3,4,5,6], PDAC patients continue to have a dismal prognosis, with a 5-year survival rate of less than 20% [7,8]. Tumor stage is the main prognostic determinant, early-stage tumors being associated with a longer survival than that of locally advanced or metastatic tumors [9]. However, rapidly evolving tumors with metastatic spread are relatively frequent, even when PDAC is diagnosed at an early stage. Differences in tumor biology probably underlie the different clinical evolution of tumors that, at diagnosis, appear similar in their anatomic extension (i.e., the same TNM characteristics).

PDAC might harbor several mutations of the cell genome, and a vast heterogeneity among tumors has been described [10,11]. Despite this complexity, some clustered recurrent mutations have allowed the classification of PDAC into four main types associated with different prognoses [12], but also the classification of neoplastic cysts with implications for the clinical decision, thus allowing patients to be spared unnecessary surgery [13]. On a ground of high genetic heterogeneity across tumors, mutations of some few genes are strongly correlated with PDAC, namely those of the *KRAS* oncogene and of *TP53*, *CDKN2A* and *SMAD4* tumor suppressor genes [14]. Among these genetic alterations, the most significantly associated with prognosis is *SMAD4* loss, found in about 60% of PDAC cases [15,16].

Although the above gene mutations alone or combined might determine the oncogenic and metastatic phenotype, molecules involved in regulating gene expression, namely microRNA (miRNAs), long non-coding RNA and circular RNA, have been the focus of attention in the last few decades due to their potential role in understanding tumor biology, and also in making the diagnosis, establishing the prognosis and providing tailored therapy [17,18,19,20]. miRNAs are small (20–25 nucleotide sequences) non-coding single-stranded RNAs that, by binding complementary mRNA sequences, might repress mRNA translation or enhance mRNA degradation, thus contributing to the regulation of a wide range of cellular activities. Any single mRNA molecule might be regulated by several miRNAs, and any miRNA might regulate several different mRNA molecules [21]. The expression of the key PDAC-associated oncogene *KRAS*, as well as that of genes encoding proteins entering in the core signaling pathways of PDAC, such as JAK/STAT, Wnt/β-catenin or TGF-β, evidences a number of regulating miRNAs with expression levels that are considered targetable by mimic or inhibitor compounds as new potential therapeutic strategies [14,18,22,23]. Moreover, miRNAs represent a great challenge for diagnosis, since they might be released by the tumor into biological fluids, such as plasma/serum or ascitic fluid [24,25], and because they are stable, thus limiting the impact of pre-analytical variability on the results. In the PDAC setting, several studies in the literature have evaluated different miRNAs panels as diagnostic and/or prognostic tools by analyzing tissue expression [26,27], serum levels [24], blood levels [28] and ascitic fluid levels [25]. New biomarkers of PDAC should aid early diagnosis and, as prognostic indices in early-stage cases, should be helpful in distinguishing between patients with long or short life expectancies, thus allowing a more tailored therapeutic approach [7,29].

The aim of the present retrospective study was, using microarray analysis of 2.549 miRNAs, to identify in sera those miRNAs able to diagnose early PDAC and to predict long or short survival after curative surgery.

## 2. Materials and Methods

### 2.1. Patients and Samples

The exploratory cohort comprised archival sera of 15 PDAC patients selected from a large retrospective series for microarray analysis. To ascertain the effects of survival on results, the initial selection criteria were: available data on survival allowing classification into two groups with short (<one year) and long (>two years) survival, tumor stage, tumor site, treatment, metabolic and biochemical characteristics being common to the two groups. On these bases the following criteria were established: (1) early-stage tumor; (2) tumor of the pancreas head; (3) R0 surgical treatment; (4) presence of diabetes mellitus or reduced glucose tolerance; (5) absence of jaundice, considering bilirubin as a potential interfering compound. Since only six cases met these criteria, we extended criteria to allow the potential presence of jaundice and tumor stage III besides stages I and II.

In addition, archival sera of four subjects with mild gastritis but without any present or previous evidence of cancer (2 males, 2 females, age range 62–74 years), were used as controls.

Another independent validation cohort of patients was selected from our archive. This included a reference group of 8 patients with mild gastritis without any present or previous evidence of cancer (7 males, 1 female, age range 42–79 years) and a series of 24 PDAC patients (6 males, 18 females, age range 48–83 years). PDAC stage was: IA = 4 patients; IIB = 11 patients; III = 8 patients; IV = 1 patient.

All sera were stored in aliquots at −80 °C and never subjected to thaw–freeze cycles before the analyses. 

### 2.2. RNA Extraction

Total RNA, including miRNAs, was extracted from the archival sera (600 μL) using the miRNeasy Serum/Plasma Advanced Kit (Qiagen, Hilden, Germany) according to the manufacturer’s instructions. miRNAs were quantified by Qubit microRNA assay kit (Thermo Fisher Scientific, Waltham, MA, USA). The RNA amount ranged from 3.5 to 18.9 ng/μL.

RNA integrity, and the content of miRNAs (%) in each sample were assessed by capillary electrophoresis with the Small RNA LabChip using the Agilent Bioanalyzer 2100 (Agilent Technologies, Santa Clara, CA, USA). Only samples with a quantity of ≥4000 pg were used for microarray analysis.

### 2.3. MicroRNA Expression Profiling

miRNAs expression profiles were obtained using “Agilent SurePrintG3 Human miRNA v.21 (8x60K)” microarray (Agilent Technologies, Santa Clara, CA, USA), which allows the detection of 2549 known human (miRBase Release 21.0) and 76 viral miRNAs (GEO Platform N. GPL24741).

Every slide contains eight individual microarrays, with 60,000 features each, including 2164 controls, used to estimate fluorescence background and background variance. Each miRNA was targeted by 16 to 20 array-probes of different sizes. Total RNA (4000 pg) was labelled with pCp Cy3, according to the Agilent’s protocol, and unincorporated dyes were removed with MicroBioSpin6 columns (BioRad, Hercules, CA, USA) [30]. Probes were hybridized at 55 °C for 22 h using the Agilent’s hybridization oven, which is suitable for bubble-mixing and microarray hybridization processes. Slides, washed by Agilent Gene expression wash buffers 1 and 2, were examined using an Agilent microarray scanner (model G2565CA) at 100% and 5% sensitivity settings. Agilent Feature Extraction software version 12.0.0.7 was used for image analysis of miRNA expression arrays. Raw miRNA data are available in the U.S. National Centre for Biotechnology Information Gene Expression Omnibus (GEO, http://www.ncbi.nlm.nih.gov/geo (accessed on 16 March 2021)) database with the Accession N. GSE168996.

### 2.4. Statistical and Functional Analysis of miRNA Expression Data

Inter-array normalization of miRNA expression levels was performed with cyclic Lowess for miRNA [31], the average of replicates being used. Feature Extraction software (Agilent Technologies, Santa Clara, CA, USA) was employed to obtain spot quality measures for evaluating the quality and the reliability of the hybridization. In particular, the flag “glsFound” (set to 1 if the spot had an intensity value significantly different from that of the local background, 0 otherwise) was used to filter out unreliable probes: a flag equal to 0 was noted as “not available” (NA). In order to make a robust and unbiased statistical analysis, probes with a high proportion of NA values were removed from the dataset. NA (47%) was used as a threshold in the filtering process, a total of 142 available human miRNAs being obtained. Cluster analysis with the average linkage method and Pearson correlation and profile similarity searches were performed with the Multi Experiment Viewer 4.9.1 (tMev) of the TM4 Microarray Software Suite [32]. All heat maps were obtained by morpheus software1 (https://software.broadinstitute.org/morpheus (accessed on 24 June 2021), Broad Institute, Cambridge, MA, USA).

Differentially expressed miRNAs were identified with two-class Significance Analysis of Microarray (SAM) algorithm [33] with default settings. SAM, which uses a permutation-based multiple testing algorithm, associates a variable false discovery rate (FDR) with the significant genes. FDR, which refers to the percentage of error that can occur in the identification of the statistically significant differentially expressed miRNAs in multiple comparisons, can be manually adjusted.

miRTarBase [34], a current and curated collection of miRNA-target interactions with experimental support, was used to predict target genes of differentially expressed miRNAs between PCDAC patients and controls. Biological pathway analysis of putative target genes was performed using DAVID v. 6.8 [35], which combines web tools such as gene functional classification, functional annotation chart or clustering and functional annotation table.

### 2.5. Reverse Transcription and Quantitative PCR (qRT-PCR) of miRNAs

Before starting the miRNA extraction procedure, 1 μL of Qiagen RNA Spike-In mix (UniSp2, UniSp4, and UniSp5) was added to 200 μL of serum (starting volume) to control the sample-to-sample variation in the RNA isolation procedure. Subsequently, total RNA, including miRNAs, was extracted using the miRNeasy Serum/Plasma Advanced Kit (Qiagen) according to the manufacturer’s instructions. cDNA was synthesized using the miRCURY LNA RT Kit (Qiagen, Hilden, Germany) starting from 75 ng of total RNA in 10 μL with the addition of 1 μL of UniSp6 as exogenous miRNA spiked-in control. PCR was performed in a 10 μL volume containing 5 μL 2x miRCURY SYBR GREEN Master Mix (Qiagen, Hilden, Germany), 1 μL cDNA, and 1 μL of one of the following miRCURY LNA PCR primer sets (Qiagen): *hsa-miR-4516* (ID YP02112882), *hsa-miR-6089* (ID YP02109969), and *UniSp4* (ID YP00203953). The qPCR reactions were performed in an ABI7500 Real-Time PCR System (Applied Biosystems, Foster City, CA, USA). The qPCR cycling conditions were 95 °C for 2 min and 40 cycles (95 °C for 10 s and 56 °C for 1 min). Three replicates of each sample and control were amplified for each real-time PCR reaction. The relative expression levels between samples were calculated using the comparative delta Ct (threshold cycle number) method (2^−ΔΔCt^), implemented in the 7500 Real Time PCR System software.

### 2.6. Statistical Analysis of Data

The non-parametric Kruskal–Wallis test, receiver operating characteristic curves (ROC) and binary logistic regression analyses were performed by Stata 13.1 (StataCorp, 4905 Lakeway Drive, TX, USA).

## 3. Results

### 3.1. Serum miRNA Expression Signatures for Early PDAC Diagnosis

Table 1 shows the clinical and biochemical characteristics of the 15 PDAC patients of the exploratory cohort, seven being short-term (<14 months), and eight long-term (>22 months) survivors. All 15 patients had a confirmed histological diagnosis of PDAC with grading, and all underwent radical surgery (R0) and had diabetes mellitus or reduced glucose tolerance diagnosed within 5 months (concurrent) or more than 5 months prior to PDAC diagnosis.

We analysed the expression of serum miRNAs in the 15 PDAC patients and in four controls using a 2549-miR microarrays platform. At SAM two-class analysis, 76 differentially expressed miRNAs were found among controls and PDAC, with an FDR of 0.4%. Interestingly, the large majority (50/76 samples, 66%) of differentially expressed miRNAs were downregulated in the sera of PDAC patients with respect to controls (Appendix A). A total of 7 (*hsa-miR-6126, hsa-miR-2392, hsa-miR-4327, hsa-miR-939-5p, hsa-miR-4655-3p, hsa-miR-371b-5p, hsa-miR-3135b*) out of the 76 miRNAs were undetectable in controls, but were expressed, albeit at low levels, only in PDAC patients’ sera.

Microarray data of any patient and control were then used for further analyses. We performed binary logistic regression analysis considering PDAC diagnosis as a dependent variable, and any of the 76 differentially expressed miRNAs between controls and PDAC (Appendix A), with age and gender, as predictor variables. Six out of the seventy-six miRNAs were significantly correlated with PDAC diagnosis, independently from age and gender: *hsa-miR-6089* (χ^2^ = 8.58, *p* = 0.050), *hsa-miR-4466* (χ^2^ = 9.19, *p*= 0.047), *hsa-miR-6821-5p* (χ^2^ = 12.05, *p* = 0.048), *hsa-miR-4669* (χ^2^ = 7.62, *p* = 0.050), *hsa-miR-1202* (χ^2^ = 6.42, *p* = 0.048), *hsa-miR-574-3p* (χ^2^ = 9.03, *p* = 0.049). The same binary logistic regression analysis performed with CA 19-9, age, and gender as predictors, was not significant (χ^2^ = 5.81, *p* = 0.232).

To further evaluate whether miRNAs outperform with respect to CA 19-9 in the early detection of PDAC, we compared receiver operating characteristic (ROC) curves of the 76 above reported miRNAs with the ROC curve of CA 19-9 considering the two groups of controls and PDAC. Those miRNAs showing an area under the ROC curve higher than that of CA 19-9 (0.8083 ± 0.0984, 95%CI: 0.61551–1.00000) are reported in Table 2.

### 3.2. miRNA Expression Profiles and PDAC Survival Rates

The circulating miRNAs expression profiles of controls were compared with those from patients with long (eight samples) and with short (seven samples) survival in order to identify miRNA expression signatures associated with different prognoses.

An unsupervised hierarchical clustering analysis, by using the list of differentially expressed genes between PDAC patients and controls, enabled the clear separation of controls and PDAC patients with long (Figure 1a) and short (Figure 1b) survival, respectively.

A total of 71 and 80 differentially expressed miRNAs between long-term and short-term survivors with respect to the same controls were identified by SAM two-class analysis with an FDR < 3%. We observed an overall downregulation of miRNAs both in long-term (47/71, 66% differentially expressed miRNAs) and short-term (56/80, 70% differentially expressed miRNAs) survivors with respect to controls (Appendix A). Among the above-mentioned miRNAs expressed in sera of PDAC patients, but not in those of controls, *hsa-miR-3135b* was detected only in patients with a short survival, whereas *hsa-miR-6126* and *hsa-miR-486-5p* were specifically observed in long-term survivors.

A SAM two-class analysis to compare short-term and long-term PDAC survivors to identify prognosis-specific signatures of circulating miRNAs, enabled the identification of 71 differentially expressed miRNAs with an extremely high FDR (62%), thus indicating that circulating miRNA expression profiles do not allow the correct stratification of PDAC patients according to their different prognoses. However, although almost all differentially expressed miRNAs (69/71, 97%) did not attain statistical significance, they were downregulated in PDAC patients with a poor prognosis with respect to those with a longer survival (Appendix A).

We performed binary logistic regression analysis considering survival of PDAC patients (long or short) as dependent variable, and miRNAs, age, and gender as predictor variables. In the analyses entered the 76 miRNAs differentially expressed between controls and PDAC (Appendix A), and the series of 71 miRNAs differentially expressed between PDAC patients with long and short survival (Appendix A). None of the evaluated miRNA reached the statistical significance.

To further ascertain whether serum miRNAs could predict a PDAC disease that is more likely with an adverse outcome, we evaluated the presence, if any, of an association between miRNAs and lymph node metastases, which was not found for CA 19-9 (χ^2^ = 4.282, *p* = 0.1175). All 76 differentially expressed miRNAs between PDAC and controls were analyzed by Kruskal–Wallis rank test comparing controls with PDAC patients with or without lymph-node metastases. This resulted in 11 significant miRNAs, reported in Table 3.

Multiple comparison reached a significant difference for eight miRNAs, which expression levels are depicted in Figure 2.

### 3.3. Validation of Selected miRNAs in an Independent Cohort of PDAC Patients Using qRT-PCR

To validate the microarray expression data with an independent technique, we performed a qRT-PCR of two miRNAs (*hsa-miR-4516* and *hsa-miR-6089*) upregulated in PDAC with respect to controls, in a novel validation cohort of 24 PDAC patients and 8 controls. Figure 3 shows the individual results of these two miRNAs, which was significantly different between PDAC and controls for *hsa-miR-4516* (Kruskal–Wallis rank test: *p* = 0.0330), not for *hsa-miR-6089* (*p* = 0.4862), although values tended to be higher in PDAC than in controls.

We then ascertained whether these two miRNAs were correlated with survival, which ranged from 3 months to 10 years, with a median of 23 months. Cox regression analysis (Table 4) was performed, including the two evaluated miRNAs, age and gender, tumor stage, CA 19-9 and hemoglobin as predictors. Survival was significantly correlated with tumor stage, but also with *hsa-miR-4516* and hemoglobin, not with CA 19-9.

### 3.4. Identification and Functional Analysis of Putative miRNAs Targets

miRTarBase (http://miRTarBase.mbc.nctu.edu.tw (accessed on 16 June 2021)) was used to predict putative targets of the 76 differentially expressed miRNAs between controls and PDAC patients (long and short survival). A list of 11.129 miRNA-target interactions (Appendix A) were identified for a total of 5.183 putative target genes. In fact, a single miRNA has hundreds of putative targets because it has pleiotropic effects on different target genes based on the biological condition being explored. KEGG pathway analysis of putative targets was performed by using the DAVID web tool to identify the main biological pathways in which are involved the differentially expressed miRNAs among PDAC patients and controls. Among the 53 most enriched KEGG pathways (EASE < 0.05, Appendix A), it is interesting to note that we found a large majority of putative targets involved in biological processes associated with cancer. The “Pancreatic cancer” pathway (Table 5) was one of the most enriched and this confirms the involvement of differentially expressed serum miRNAs in PDAC pathogenesis and/or progression. We also observed a statistically significant enrichment of target genes in biological processes that are considered highly important for cancer progression and dissemination, such as the “Wnt signalling pathway” (Appendix A), the “p53 signalling pathway” (Appendix A) and the “TGF-beta signalling pathway” (Appendix A).

## 4. Discussion

PDAC is the only cancer type with a survival rate that has not improved in the last 40 years [8]. Since most patients are diagnosed when they have metastases, they can only be considered candidates for palliative treatment. Radical surgery for patients with locally confined tumors might not provide long-term survival, the median survival interval following diagnosis ranging from eight to ten months and early tumor relapse occurring in most patients [29]. Several factors affect the survival rates, such as cancer type, tumor stage at diagnosis, treatment modality, age, sex, overall health, lifestyle and differences in healthcare systems [36,37].

Although survival duration is mainly related to tumor stage, both long-term and short-term survivals have been described in patients diagnosed with early-stage tumors [29,36,37]. Recent studies have reported the prognostic utility of circulating miRNAs profiling in several malignancies due to their altered expression during tumorigenesis and their stability in body fluids [17,18]. In sera, miRNAs could be either entrapped in exosomes or they could circulate bound to proteins, such as Ago2 [38]. Although circulating exosomes are important vehicles of tumor-derived miRNAs and a potential source for biomarker identification, the overall RNA yield that could be obtained after exosomes isolation is generally lower than that from pooled sera, thus compromising test sensitivity [39]. For this reason, we choose to analyse whole sera.

In the present retrospective study, we performed whole miRNAs microarray analysis in the sera of cancer-free controls and of selected early-stage PDAC patients with different survival rates (<14 months or >22 months after curative surgery) in order to establish whether miRNAs measurement can enable early disease diagnosis and predict survival in a minimally invasive way. Unsupervised cluster analysis allowed a distinction between PDAC patients and controls. A number of 76 miRNAs were identified as differently expressed between PDAC patients (short and long-term survivors) with respect to controls, and the large majority of them were downregulated in PDAC patients with respect to controls, in line with findings reported in the current literature. In fact, the overall downregulation of miRNAs, emerging as a common hallmark of cancer, contributes to the malignant phenotype and a poor prognosis [40]. Our results not only corroborate this assumption, but also indicate that this is an early and detectable finding, since our patients’ series comprised only early-stage PDAC. Furthermore, seven circulating miRNAs (*hsa-miR-6126*, *hsa-miR-2392*, *hsa-miR-4327*, *hsa-miR-939-5p*, *hsa-miR-4655-3p*, *hsa-miR-371b-5p*, *hsa-miR-3135b*), expressed in low levels, were found only in the sera of PADC patients and not in controls, thus suggesting their potential role as an emerging class of suitable biomarkers for this cancer type.

Further studies are required to investigate the effect of these selected miRNAs, which were until now poorly evaluated in the PDAC setting, although some of them have been studied in other cancer types. Our findings are in agreement with those of Mazza et al., who demonstrated that *hsa-miR-6126* was one of the most significantly increased miRNAs in the plasma of PDAC patients [41], and the same *hsa-miR-6126* has been found in the tumor tissues of five patients with stage II colon cancer but not in cancer-free tissues [42]. *hsa-miR-939-5p*, of potential interest for its role in invasive cancer, is reportedly upregulated in lung adenocarcinomas [43], hepatocellular carcinomas [44,45], ovarian cancer tissue [46] and in the promotion of blood vessel invasion in breast cancer [47]. The recent study by Shen et al. [48] shed some light on the role of this *hsa-miR-939-5p* in PDAC biology and prognosis. By studying human pancreatic cancer tissues and cell lines, the authors demonstrated that high expression levels of this miRNA are associated with a poor prognosis in patients, while promoting in vitro tumor cell migration and invasion by targeting the Rho GTPase, activating protein 4.

Another series of six miRNAs was found to be significantly and independently correlated with early PDAC, namely *hsa-miR-6089*, *hsa-miR-4466*, *hsa-miR-6821-5p*, *hsa-miR-4669*, *hsa-miR-1202* and *hsa-miR-574-3p*. Among these, *hsa-miR-6821-5p* was associated with the best ROC curve area in distinguishing controls from early PDAC, supporting its potential role in early diagnosis. This result fits well with findings of Keller et al. [49], who demonstrated that *hsa-miR-6821-5p* is one of the most deregulated miRNAs in serum for decades prior lung, breast, or colon cancer diagnosis. The same study emphasized *hsa-miR-4687-3p* and *hsa-miR-574-3p* as potential pre-diagnostic markers and, in agreement, we identified these miRNAs among those most significant in distinguishing early PDAC from controls. All the above-described miRNAs outperformed with respect to the established PDAC biomarker CA 19-9, which levels did not significantly differ between controls and patients. This finding is not surprising, considering its low sensitivity for early tumor diagnosis [50].

When miRNA expression data of PDAC patients were analysed in order to identify a miRNA transcriptional signature characterizing early-stage PDAC patients with different prognoses, unsupervised cluster analysis did not enable a distinction between short-term and long-term survivors, but SAM two-class analysis (high FDR: 62.5%) allowed the identification of 71 differentially expressed miRNAs. The majority of survival-related differentially expressed miRNAs were downregulated in short-term survival with respect to long-term survival patients, thus confirming the association between cancer progression and miRNA downregulation, as reported above [40].

We first focused our attention on the three miRNAs that were expressed exclusively in the sera of PDAC patients with long-term (*hsa-miR-486-5p* and *hsa-miR-6126*) or short-term (*hsa-miR-3135b*) survival, but not in controls.

The finding of *hsa-miR-486-5p* expression exclusively in long-term survival patients supports the hypothesis that tumor progression determines a progressive decline for this miRNA expression, as occurs in colorectal cancer [51]. While no data in the literature report an association between *hsa-miR-3135b* and cancer progression and/or survival, *hsa-miR-6126* has been recently identified by Mazza et al. in PDAC plasma with levels increasing with metastases, but not correlated with survival [41]. The differences between our results and the results of Mazza et al. [41] might depend on the different characteristics of the patients studied: our series was mainly represented by stage I-II, while the series of Mazza et al. by stage III-IV PDAC cases.

To obtain further insights into the association between serum miRNAs and outcome, we verified whether miRNAs were correlated with the presence or absence of lymph-node metastases, known to be associated with worse outcome [52]. Eight miRNAs were identified among the most significant (Figure 2). Few data are present in the literature on these miRNAs, but *hsa-miR-1202*, which decreased in the presence of lymph-node metastases, was described to similarly decline with the progression of cervical cancer in tissue samples [53] and it was shown to exert an anti-tumor effect in vitro on HCC cells lines [54]. The *hsa-miRNA-4669*, which also declines in the presence of lymph-node metastases, was also described to diminish in sera of colorectal cancer patients [55].

One upregulated miRNA, namely hsa-miR-4516, was confirmed by qRT-PCR in an independent PDAC patients’ validation cohort to have increased expression levels in PDAC with respect to controls. This miRNA was also an independent predictor of survival, which was correlated as expected with tumor stage, but also with hemoglobin levels, not with CA 19-9. Furthermore, the putative targets of this miRNA were statistically significantly enriched in the target’s prediction analysis performed with miRTarBase, and they are involved in key pathways of cancer progression and dissemination such as the “Wnt signalling pathway” (*CCND2*, *RAC3*, *TP53*, *WNT8B*, *FBXW11*, *SOX17*, *TBL1XR1*) and the “p53 signalling pathway” (*CDKN1A*, *SFN*, *TP53*, *RPRM*).

Overall, our results support the importance of exploring the use of liquid biopsies in PDAC patients, managing the clinical utility of circulating miRNAs as novel diagnostic and prognostic biomarkers for this disease.

The main limitation of our study is related to the few numbers of the studied patients in both the exploratory and validation cohorts. This limitation derives from the choice to study early tumors with short or particularly long survival that imposed strict selection criteria (cases of early-stage cancer, without jaundice, with diabetes or reduced glucose tolerance, and with survival data including long-term survival). Considering that at diagnosis, PDAC patients with stage I–II are less than 20% [56], this left few cases within a large retrospective cohort meeting the requirements. On the other hand, our findings were not biased by tumor stage and allowed to consider the identified circulating miRNAs as early biomarkers.

## 5. Conclusions

In conclusion, the findings made in the present study suggest that new circulating miRNAs are a potentially useful biomarker for making an early diagnosis of PDAC and for establishing the prognosis. These miRNAs include *hsa-miR-3135b*, *hsa-miR-6126, hsa-miR-486-5p, hsa-miR-6821-5p, hsa-miR-4669* and *hsa-miR-4516* that, after future and independent validation, could be measured in the sera of PDAC patients, thus enabling improved management and prolonging survival.

## Figures and Tables

**Figure 1 biomedicines-09-00845-f001:**
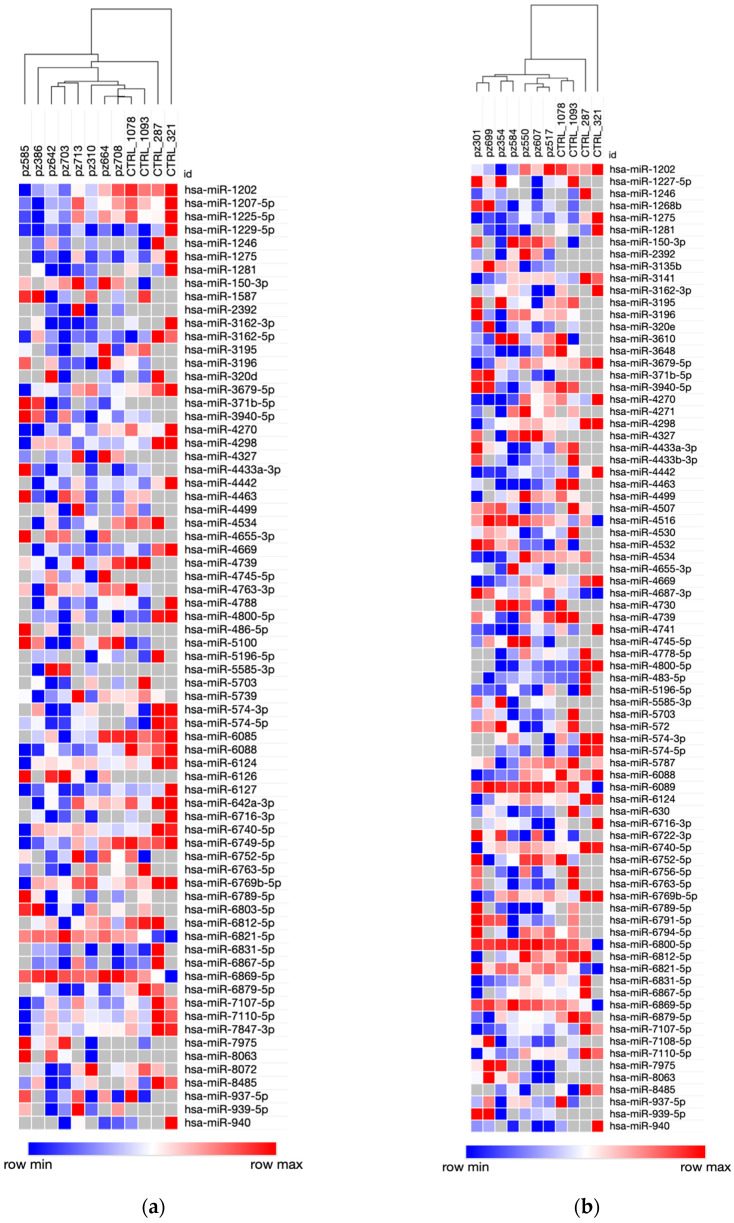
Cluster analysis of differentially expressed miRNAs between PDAC patients and controls. Heat maps representing differentially expressed miRNAs of two different comparisons: long survival patients (**a**) and short survival (**b**) patients with respect to controls. Both dendrograms show the capability of circulating miRNA signatures to clearly separate PDAC patients with respect to controls. A color-coded scale for the normalized expression values is used: red and blue represent high and low expression levels. A complete list of differentially expressed genes identified by SAM two-class algorithm is provided in the Appendix A.

**Figure 2 biomedicines-09-00845-f002:**
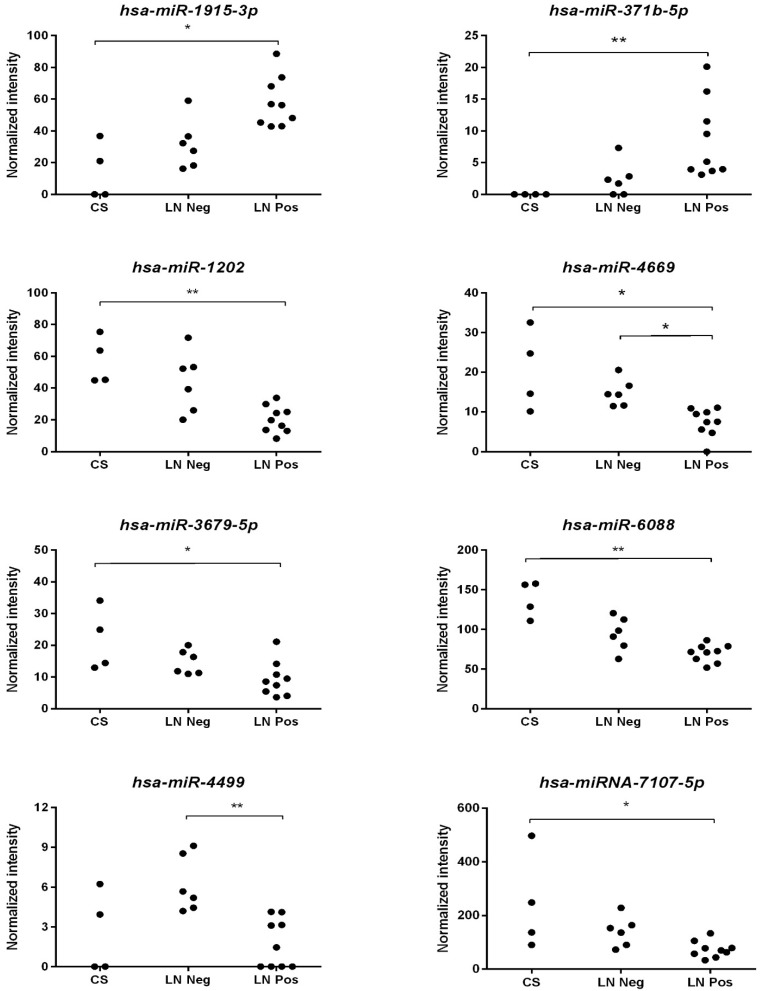
Expression levels of miRNAs significantly predicting adverse outcome. Normalized intensity values of miRNAs expression levels among controls (CS), PDAC patients with (LN pos) and without (LN neg) lymph node metastases have been compared. *: *p* < 0.05; **: *p* < 0.005.

**Figure 3 biomedicines-09-00845-f003:**
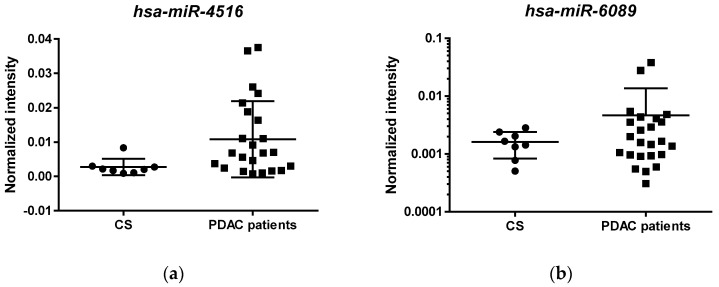
Relative quantification of *hsa*-*miR-4516* (**a**) and *hsa*-*miR-6089* (**b**) serum levels in an independent cohort of PDAC patients. The relative serum expression levels of *hsa*-*miR-4516* and *hsa*-*miR-6089* measured in 8 controls (CS) and in 24 PDAC patients are shown.

**Table 1 biomedicines-09-00845-t001:** Clinical and biochemical characteristics of PDAC patients included in the study.

Patient Nr.	Age	Gender	G	T	N	M	Stage	Survival (months)	Diabetes	CA 19-9kU/L	Total Bilirubin μmol/L	P-Glucose mmol/L	Survival
713	57	M	2	2	0	0	IB	113	DM (C)	131	10	5.0	L
708	70	M	1	3	0	0	IIA	86	DM (P)	1	8	10.4	L
386	59	M	2	3	1	0	IIB	75	RGT (C)	20	92	6.1	L
664	79	F	2	3	1	0	IIB	56	RGT (C)	na	39	4.7	L
642	79	M	2	3	0	0	IIA	45	RGT (C)	92	39	7.4	L
310	78	F	2	3	1	0	IIB	29	DM (P)	6969	135	8.8	L
703	72	F	2	3	1	0	IIB	25	DM (P)	5309	12	8.9	L
585	63	M	2	3	1	0	IIB	23	DM (P)	435	13	8.2	L
584	58	F	3	3	1	0	IIB	13	RGT (C)	106	81	5.2	S
354	63	M	2	4	1	0	III	11	DM (C)	519	94	5.9	S
301	65	M	3	3	1	0	IIB	11	RGT (C)	48	65	7.1	S
699	67	F	2	3	1	0	IIB	9	DM (C)	30	22	6.4	S
517	61	F	3	4	0	0	III	8	DM (C)	212	9	6.2	S
607	68	M	3	3	0	0	IIA	6	RGT (C)	29	7	5.3	S
550	84	F	2	3	0	0	IIA	2	DM (P)	105	39	8.1	S

Diabetes: DM = Diabetes Mellitus; RGT = Reduced Glucose Tolerance; C = concurrent diabetes diagnosis (≤5 months); P = previous diabetes diagnosis (>5 months); Survival: L = Long (>22 months); S = Short (<14 months).

**Table 2 biomedicines-09-00845-t002:** Serum miRNAs and early PDAC diagnosis. miRNAs showing an area under the ROC curve (AUC) higher than that of CA 19-9 in distinguishing controls from PDAC are reported. SE: standard error; 95% CI: 95% confidence interval; Increase/Decrease: miRNA levels increased or decreased in PDAC with respect to controls. Bold face: miRNAs significantly correlated with PDAC diagnosis at binary logistic regression analysis.

		ROC		Asymptotic Normal		Increase/Decrease
	Obs	AUC	SE	95% Conf. Interval		
*hsa-miR-7110-5p*	19	0.8167	0.1360	0.55014	1.00000	Decrease
*hsa-miR-3135-b*	19	0.8333	0.0630	0.70987	0.95680	Increase
** *hsa-miR-4669* **	19	0.8333	0.1346	0.56959	1.00000	Decrease
*hsa-miR-7107-5p*	19	0.8333	0.1225	0.59329	1.00000	Decrease
** *hsa-miR-574-3p* **	19	0.8333	0.1225	0.59329	1.00000	Decrease
*hsa-miR-1275*	19	0.8333	0.1188	0.60049	1.00000	Decrease
** *hsa-miR-4466* **	19	0.8500	0.1351	0.58529	1.00000	Increase
*hsa-miR-3679-5p*	19	0.8500	0.1055	0.64316	1.00000	Decrease
*hsa-miR-2392*	19	0.8667	0.0591	0.75085	0.98249	Increase
*hsa-miR-4655-3p*	19	0.8667	0.0591	0.75085	0.98249	Increase
** *hsa-miR-6089* **	19	0.8667	0.1196	0.63230	1.00000	Increase
*hsa-miR-5100*	19	0.8667	0.0970	0.67646	1.00000	Increase
*hsa-miR-6749-5p*	19	0.8667	0.0858	0.69858	1.00000	Decrease
*hsa-miR-4687-3p*	19	0.8833	0.0922	0.70258	1.00000	Increase
*hsa-miR-1915-3p*	19	0.8833	0.0922	0.70258	1.00000	Increase
*hsa-miR-6125*	19	0.8833	0.1043	0.67896	1.00000	Increase
** *hsa-miR-1202* **	19	0.8833	0.0812	0.72426	1.00000	Decrease
*hsa-miR-8485*	19	0.8833	0.1043	0.67896	1.00000	Decrease
*hsa-miR-6126*	19	0.9000	0.0535	0.79524	1.00000	Increase
*hsa-miR-939-5p*	19	0.9000	0.0535	0.79524	1.00000	Increase
*hsa-miR-6800-5p*	19	0.9000	0.0796	0.74399	1.00000	Increase
*hsa-miR-4516*	19	0.9167	0.0684	0.78255	1.00000	Increase
*hsa-miR-6869-5p*	19	0.9167	0.0891	0.74206	1.00000	Increase
*hsa-miR-6850-5p*	19	0.9167	0.0746	0.77041	1.00000	Increase
*hsa-miR-4327*	19	0.9333	0.0454	0.84430	1.00000	Increase
*hsa-miR-371b-5p*	19	0.9333	0.0454	0.84430	1.00000	Increase
** *hsa-miR-6821-5p* **	19	0.9667	0.0403	0.88761	1.00000	Increase

**Table 3 biomedicines-09-00845-t003:** The Table reports results of Kruskal–Wallis rank test considering lymph node metastases as variable defining groups and any individual miRNA as predictor variable.

	χ^2^	*p*
*hsa-miR-1202*	10.861	0.0044
*hsa-miR-3679-5p*	8.061	0.0178
*hsa-miR-6088*	11.782	0.0028
*hsa-miR-6791-5p*	6.027	0.0491
*hsa-miR-1915-3p*	10.861	0.0044
*hsa-miR-371b-5p*	11.982	0.0025
*hsa-miR-4669*	12.430	0.0020
*hsa-miR-4499*	9.682	0.0079
*hsa-miR-4442*	8.219	0.0164
*hsa-miR-7107-5p*	9.482	0.0087
*hsa-miR-4800-5p*	6.685	0.0353

**Table 4 biomedicines-09-00845-t004:** Cox regression analysis in the validation cohort. Dependent variable: PDAC patients’ survival.

Predictors	HR	SE	z	*p*	95% CI	
Sex	11.2116	15.25747	1.78	0.076	0.7785689	161.4499
Age	1.088652	0.0546704	1.69	0.091	0.9866048	1.201255
*hsa-miR-4516*	1.59 × 10^−46^	7.36 × 10^−45^	−2.28	0.023	6.58 × 10^−86^	3.85 × 10^−07^
*hsa-miR-6089*	2.91 × 10^34^	1.24 × 10^36^	1.86	0.063	0.0129917	6.53 × 10^70^
CA 19-9	1.000101	0.0000555	1.81	0.070	0.9999919	1.000209
*Hemoglobin*	0.9515861	0.0202986	−2.33	0.020	0.9126218	0.9922141
Stage	20.64211	22.01492	2.84	0.005	2.552365	166.9418

**Table 5 biomedicines-09-00845-t005:** List of miRNA-target interactions enriched in the “Pancreatic cancer” pathway. Bold face: the interactions between *miR-4516* (validated in qRT-PCR) and its target genes.

Entrez ID	Symbol	Gene Name	miRNA	Log2 (PDAC Patients/Control)	Experimental Evidence
208	*AKT2*	*AKT serine/threonine kinase 2*	*hsa-miR-2861*	0.60	Luciferase reporter assay; Western blot
9459	*ARHGEF6*	*Rac/Cdc42 guanine nucleotide exchange factor 6*	*hsa-miR-6127*	−1.90	PAR-CLIP
598	*BCL2L1*	*BCL2 like 1*	*hsa-miR-6127*	−1.90	PAR-CLIP
598	*BCL2L1*	*BCL2 like 1*	*hsa-miR-7110-5p*	−1.11	PAR-CLIP
598	*BCL2L1*	*BCL2 like 1*	*hsa-miR-4739*	−0.98	PAR-CLIP
598	*BCL2L1*	*BCL2 like 1*	*hsa-miR-5787*	−0.74	PAR-CLIP
598	*BCL2L1*	*BCL2 like 1*	*hsa-miR-6879-5p*	−0.66	PAR-CLIP
598	*BCL2L1*	*BCL2 like 1*	*hsa-miR-6756-5p*	−0.30	PAR-CLIP
598	*BCL2L1*	*BCL2 like 1*	*hsa-miR-6752-5p*	0.02	PAR-CLIP
598	*BCL2L1*	*BCL2 like 1*	*hsa-miR-6791-5p*	0.09	PAR-CLIP
598	*BCL2L1*	*BCL2 like 1*	*hsa-miR-371b-5p*	null	PAR-CLIP
598	*BCL2L2*	*BCL2 like 1*	*hsa-miR-630*	−1.34	Immunoblot;Luciferase reporter assay;qRT-PCR
595	*CCND1*	*cyclin D1*	*hsa-miR-574-5p*	−2.63	PAR-CLIP
595	*CCND1*	*cyclin D1*	*hsa-miR-3648*	−1.98	PAR-CLIP
595	*CCND1*	*cyclin D1*	*hsa-miR-5196-5p*	−1.69	PAR-CLIP
595	*CCND1*	*cyclin D1*	*hsa-miR-7107-5p*	−1.27	PAR-CLIP
595	*CCND1*	*cyclin D1*	*hsa-miR-6867-5p*	−0.84	PAR-CLIP
595	*CCND1*	*cyclin D1*	*hsa-miR-3940-5p*	0.01	qRT-PCR;Western blot
595	*CCND1*	*cyclin D1*	*hsa-miR-2861*	0.60	Luciferase reporter assay;Western blot
595	*CCND1*	*cyclin D1*	*hsa-miR-2392*	null	PAR-CLIP
1019	*CDK4*	*cyclin dependent kinase 4*	*hsa-miR-3135b*	null	HITS-CLIP
1021	*CDK6*	*cyclin dependent kinase 6*	*hsa-miR-6716-3p*	−2.51	PAR-CLIP
1021	*CDK6*	*cyclin dependent kinase 6*	*hsa-miR-8485*	−1.81	HITS-CLIP
1021	*CDK6*	*cyclin dependent kinase 6*	*hsa-miR-7847-3p*	−0.82	PAR-CLIP
1021	*CDK6*	*cyclin dependent kinase 6*	*hsa-miR-4534*	−0.48	PAR-CLIP
1021	*CDK6*	*cyclin dependent kinase 6*	*hsa-miR-5739*	−0.14	PAR-CLIP
1021	*CDK6*	*cyclin dependent kinase 6*	*hsa-miR-371b-5p*	null	PAR-CLIP
1869	*E2F1*	*E2F transcription factor 1*	*hsa-miR-940*	−3.46	PAR-CLIP
1869	*E2F1*	*E2F transcription factor 1*	*hsa-miR-6763-5p*	−1.09	PAR-CLIP
1869	*E2F1*	*E2F transcription factor 1*	*hsa-miR-939-5p*	null	PAR-CLIP
1870	*E2F2*	*E2F transcription factor 2*	*hsa-miR-4669*	−0.88	HITS-CLIP
1870	*E2F2*	*E2F transcription factor 2*	*hsa-miR-4433b-3p*	−0.40	PAR-CLIP
1871	*E2F3*	*E2F transcription factor 3*	*hsa-miR-6124*	−0.94	PAR-CLIP
1871	*E2F3*	*E2F transcription factor 3*	*hsa-miR-6867-5p*	−0.84	PAR-CLIP
1871	*E2F3*	*E2F transcription factor 3*	*hsa-miR-4778-5p*	−0.33	PAR-CLIP
1871	*E2F3*	*E2F transcription factor 3*	*hsa-miR-4271*	−0.19	PAR-CLIP
1950	*EGF*	*epidermal growth factor*	*hsa-miR-940*	−3.46	PAR-CLIP
1950	*EGF*	*epidermal growth factor*	*hsa-miR-4433a-3p*	−0.17	PAR-CLIP
1956	*EGFR*	*epidermal growth factor receptor*	*hsa-miR-574-3p*	−1.57	Luciferase reporter assay;Microarray;qRT-PCR;Western blot
1956	*EGFR*	*epidermal growth factor receptor*	*hsa-miR-2861*	0.60	Luciferase reporter assay;Western blot
2064	*ERBB2*	*erb-b2 receptor tyrosine kinase 2*	*hsa-miR-4270*	−0.97	HITS-CLIP
8517	*IKBKG*	*inhibitor of kappa light polypeptide gene enhancer in B-cells, kinase gamma*	*hsa-miR-6127*	−1.90	PAR-CLIP
8517	*IKBKG*	*inhibitor of kappa light polypeptide gene enhancer in B-cells, kinase gamma*	*hsa-miR-8485*	−1.81	HITS-CLIP
8517	*IKBKG*	*inhibitor of kappa light polypeptide gene enhancer in B-cells, kinase gamma*	*hsa-miR-4739*	−0.98	HITS-CLIP
3845	*KRAS*	*KRAS proto-oncogene, GTPase*	*hsa-miR-671-5p*	−0.30	PAR-CLIP
5594	*MAPK1*	*mitogen-activated protein kinase 1*	*hsa-miR-6763-5p*	−1.09	PAR-CLIP
5594	*MAPK1*	*mitogen-activated protein kinase 1*	*hsa-miR-6831-5p*	−0.91	PAR-CLIP
5594	*MAPK1*	*mitogen-activated protein kinase 1*	*hsa-miR-6088*	−0.80	PAR-CLIP
5594	*MAPK1*	*mitogen-activated protein kinase 1*	*hsa-miR-4271*	−0.19	PAR-CLIP
5594	*MAPK1*	*mitogen-activated protein kinase 1*	*hsa-miR-4433a-3p*	−0.17	PAR-CLIP
5594	*MAPK1*	*mitogen-activated protein kinase 1*	*hsa-miR-6869-5p*	1.18	PAR-CLIP
5594	*MAPK1*	*mitogen-activated protein kinase 1*	*hsa-miR-3135b*	null	HITS-CLIP
5602	*MAPK10*	*mitogen-activated protein kinase 10*	*hsa-miR-574-5p*	−2.63	HITS-CLIP
5602	*MAPK10*	*mitogen-activated protein kinase 10*	*hsa-miR-6867-5p*	−0.84	HITS-CLIP
5599	*MAPK8*	*mitogen-activated protein kinase 8*	*hsa-miR-371b-5p*	null	PAR-CLIP
5293	*PIK3CD*	*phosphatidylinositol-4,5-bisphosphate 3-kinase catalytic subunit delta*	*hsa-miR-4433a-3p*	−0.17	PAR-CLIP
5294	*PIK3CG*	*phosphatidylinositol-4,5-bisphosphate 3-kinase catalytic subunit gamma*	*hsa-miR-8485*	−1.81	PAR-CLIP
5295	*PIK3R1*	*phosphoinositide-3-kinase regulatory subunit 1*	*hsa-miR-8485*	−1.81	PAR-CLIP
5295	*PIK3R1*	*phosphoinositide-3-kinase regulatory subunit 1*	*hsa-miR-1202*	−0.94	PAR-CLIP
5296	*PIK3R2*	*phosphoinositide-3-kinase regulatory subunit 2*	*hsa-miR-3135b*	null	PAR-CLIP
5879	*RAC1*	*ras-related C3 botulinum toxin substrate 1*	*hsa-miR-574-3p*	−1.57	Luciferase reporter assay;Microarray;qRT-PCR;Western blot
5879	*RAC1*	*ras-related C3 botulinum toxin substrate 1*	*hsa-miR-6763-5p*	−1.09	PAR-CLIP
5879	*RAC1*	*ras-related C3 botulinum toxin substrate 1*	*hsa-miR-6124*	−0.94	PAR-CLIP
5879	*RAC1*	*ras-related C3 botulinum toxin substrate 1*	*hsa-miR-939-5p*	null	PAR-CLIP
5881	*RAC3*	*ras-related C3 botulinum toxin substrate 3*	*hsa-miR-5703*	−1.70	PAR-CLIP
**5881**	** *RAC3* **	*ras-related C3 botulinum toxin substrate 3*	** *hsa-miR-4516* **	1.51	PAR-CLIP
5888	*RAD51*	*RAD51 recombinase*	*hsa-miR-940*	−3.46	PAR-CLIP
5888	*RAD51*	*RAD51 recombinase*	*hsa-miR-7847-3p*	−0.82	PAR-CLIP
5888	*RAD51*	*RAD51 recombinase*	*hsa-miR-1915-3p*	0.72	HITS-CLIP
5888	*RAD51*	*RAD51 recombinase*	*hsa-miR-7975*	1.09	HITS-CLIP
5888	*RAD51*	*RAD51 recombinase*	*hsa-miR-371b-5p*	null	HITS-CLIP
5894	*RAF1*	*Raf-1 proto-oncogene, serine/threonine kinase*	*hsa-miR-4534*	−0.48	PAR-CLIP
5894	*RAF1*	*Raf-1 proto-oncogene, serine/threonine kinase*	*hsa-miR-6789-5p*	−0.27	PAR-CLIP
5970	*RELA*	*RELA proto-oncogene, NF-kB subunit*	*hsa-miR-3162-3p*	−2.72	PAR-CLIP
5970	*RELA*	*RELA proto-oncogene, NF-kB subunit*	*hsa-miR-4534*	−0.48	PAR-CLIP
4087	*SMAD2*	*SMAD family member 2*	*hsa-miR-8485*	−1.81	HITS-CLIP
4087	*SMAD2*	*SMAD family member 2*	*hsa-miR-937-5p*	−0.38	HITS-CLIP
4089	*SMAD4*	*SMAD family member 4*	*hsa-miR-574-5p*	−2.63	HITS-CLIP
4089	*SMAD4*	*SMAD family member 4*	*hsa-miR-574-3p*	−1.57	Luciferase reporter assay;qRT-PCR;Western blot
4089	*SMAD4*	*SMAD family member 4*	*hsa-miR-6867-5p*	−0.84	HITS-CLIP
4089	*SMAD4*	*SMAD family member 4*	*hsa-miR-371b-5p*	null	PAR-CLIP
6774	*STAT3*	*signal transducer and activator of transcription 3*	*hsa-miR-4270*	−0.97	PAR-CLIP
**6774**	** *STAT3* **	*signal transducer and activator of transcription 3*	** *hsa-miR-4516* **	1.51	Luciferase reporter assay;Microarray;qRT-PCR;Western blot
7040	*TGFB1*	*transforming growth factor beta 1*	*hsa-miR-574-3p*	−1.57	Luciferase reporter assay;qRT-PCR;Western blot
7046	*TGFBR1*	*transforming growth factor beta receptor 1*	*hsa-miR-6831-5p*	−0.91	HITS-CLIP
7048	*TGFBR2*	*transforming growth factor beta receptor 2*	*hsa-miR-940*	−3.46	PAR-CLIP
7048	*TGFBR2*	*transforming growth factor beta receptor 2*	*hsa-miR-574-5p*	−2.63	HITS-CLIP
7048	*TGFBR2*	*transforming growth factor beta receptor 2*	*hsa-miR-630*	−1.34	Microarray
7048	*TGFBR2*	*transforming growth factor beta receptor 2*	*hsa-miR-6867-5p*	−0.84	HITS-CLIP
7157	*TP53*	*tumor protein p53*	*hsa-miR-6127*	−1.90	PAR-CLIP
7157	*TP53*	*tumor protein p53*	*hsa-miR-5703*	−1.70	PAR-CLIP
7157	*TP53*	*tumor protein p53*	*hsa-miR-7110-5p*	−1.11	PAR-CLIP
7157	*TP53*	*tumor protein p53*	*hsa-miR-937-5p*	−0.38	PAR-CLIP
7157	*TP53*	*tumor protein p53*	*hsa-miR-6756-5p*	−0.30	PAR-CLIP
7157	*TP53*	*tumor protein p53*	*hsa-miR-4271*	−0.19	PAR-CLIP
7157	*TP53*	*tumor protein p53*	*hsa-miR-6752-5p*	0.02	PAR-CLIP
**7157**	** *TP53* **	*tumor protein p53*	** *hsa-miR-4516* **	1.51	PAR-CLIP
7422	*VEGFA*	*vascular endothelial growth factor A*	*hsa-miR-8485*	−1.81	PAR-CLIP;HITS-CLIP
7422	*VEGFA*	*vascular endothelial growth factor A*	*hsa-miR-6769b-5p*	−0.70	PAR-CLIP
7422	*VEGFA*	*vascular endothelial growth factor A*	*hsa-miR-6756-5p*	−0.30	PAR-CLIP

## Data Availability

Raw miRNA data are available in the U.S. National Centre for Biotechnology Information Gene Expression Omnibus (GEO, http://www.ncbi.nlm.nih.gov/geo (accessed on 16 March 2021)) database with the Accession N. GSE168996.

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
