# Peer review of "Serum miRNA Profiling for Early PDAC Diagnosis and Prognosis: A Retrospective Study"

_biomedicines, 2021, doi:10.3390/biomedicines9070845_

Round 1

Reviewer 1 Report

  1. What are the potential target of these miRNAs found in the study?  Do these genes related to disease progression? Maybe the author can try to predict the target genes.
  2. The unsupervised cluster process is not well described, like the miRNAs used for the clustering and how the author choose those miRNAs. Also what each row of the heat map are should be labeled.
  3. Is it possible for the author to confirm the finding in other cohort?
  4. The ROC analysis mentioned in abstract is missing in main text.

Author Response

Reviewer #1

Dear reviewer, thank you very much for the constructive criticisms made to our study “Serum miRNA profiling for early PDAC diagnosis and prognosis: a retrospective study” (biomedicines-1170598). The paper has been modified taking into account yours suggestions and it has now significantly improved.

Comment 1:

What are the potential targets of these miRNAs found in the study? Do these genes related to disease progression? Maybe the author can try to predict the target genes.

Author’s answer:

As suggested by the Reviewer we have used miRTarBase, a curated collection of miRNA-target interactions with experimental support, to predict target genes of differentially expressed miRNAs between PCDAC patients and controls (see Table S5). Furthermore, a functional enrichment analysis of the putative target genes was performed using DAVID. It is interesting to note that a large majority of these targets are involved in biological processes associated to cancer progression and dissemination such as we have described in the results (see Table 5, Table S6, S7, S8 and S9).

Comment 2:

  • The unsupervised cluster process is not well described, like the miRNAs used for the clustering and how the author choose those miRNAs. Also what each row of the heat map are should be labeled.

Author’s answer:

We agree with the reviewer that the cluster process was not well described. So, we added the following sentence in the Results to try to clarify this aspect: “An unsupervised hierarchical clustering analysis, by using the list of differentially ex-pressed genes between PDAC patients and controls, enabled the clear separation of controls and PDAC patients with long (Figure 1A) and short (Figure 1B) survival respectively.” Furthermore, we have replaced the Figure 1 with the new one in which we have represented the heat-maps of all differentially expressed miRNAs between PDAC patients with long (Figure 1A) and short (Figure 1B) survival with respect to controls. Each row corresponds to a different miRNA.

Comment 3:

  • Is it possible for the author to confirm the finding in other cohort?

Author’s answer:

To confirm the results obtained with the microarray experiments, we validated the expression levels of two circulating miRNAs (miR 4516 and miR-6089) by using qRT-PCR in a novel cohort of 24 PDAC patients and 8 controls such as we have described in the Results’ paragraph entitled: “3.3. Validation of selected miRNAs in an independent cohort of PDAC patients using qRT-PCR”.

Comment 4:

  • The ROC analysis mentioned in abstract is missing in main text.
  • Author’s answer:

Receiver characteristic curves (ROC) are shown in Table 2.

Reviewer 2 Report

The authors are describing dysregulation of circulating miRNA in serum of peripheral blood of pancreatic cancer patients.

I have several comments to this study:

  1. At this point, most studies are performed on exosomes of peripheral blood, not the entire serum.
  2. In my opinion, the study is performed on a very small sample of patients, both the exploration phase and the validation phase.
  3. what was the range of isolated RNA? what was the range of purity? what was the cutt-off?
  4. what was the amount of RNA used for cDNA? the authors said 10ul but I am wondering about the exact amount

Author Response

Reviewer #2

Dear reviewer, thank you very much for the constructive criticisms made to our study “Serum miRNA profiling for early PDAC diagnosis and prognosis: a retrospective study” (biomedicines-1170598). The paper has been modified taking into account yours suggestions and it has now significantly improved.

Comment 1:

At this point, most studies are performed on exosomes of peripheral blood, not the entire serum.

  • Author’s answer:

We agree with this reviewer. Circulating exosomes are important vehicles of tumor derived miRNAs and a potential source for biomarker identification. However, the overall RNA yield that could be obtained after exosomes isolation is generally lower than that from pooled sera thus compromising test sensitivity. For this reason we choose to analyse whole sera and this is now clearly defined in the first part of the discussion.

Comment 2:

In my opinion, the study is performed on a very small sample of patients, both the exploration phase and the validation phase.

  • Author’s answer:

We agree with this reviewer that our patients’ series in both the exploratory and validation cohorts are few. This is the main limitation of our study.  This limitation derives from the choice to study early tumors with short or very long survival that imposed strict selection criteria (cases of early-stage cancer, without jaundice, with diabetes or reduced glucose tolerance, and with survival data including long term-survival).  At diagnosis PDAC is in the majority of the cases at an advanced stage being patients with stage I-II very rare. Therefore few cases within a large retrospective cohort meeting the requirements were available. This is now cleary stated in the discussion as limitation of the study.

Comment 3:

What was the range of isolated RNA? what was the range of purity? what was the cut-off?

  • Author’s answer:

As suggested by the Reviewer all this information were added in the paragraph entitled “2.2. RNA extraction” in Materials and Methods section.

The RNA amount ranged from 3.5 to 18.9 ng/ml and only samples with a quantity of ≥4000 pg were used for microarray analysis.

Comment 4:

What was the amount of RNA used for cDNA? the authors said 10 ul but I am wondering about the exact amount.

  • Author’s answer:

As suggested by the Reviewer, this information was added in the paragraph entitled “2.5. Reverse Transcription and Quantitative PCR (qRT-PCR) of miRNAs” in Materials and Methods section:

cDNA was synthesized using the miRCURY LNA RT Kit (Qiagen) starting from 75 ng of total RNA in 10 μl with the addition of 1 μl of UniSp6 as exogenous miRNA spiked-in control.

Round 2

Reviewer 2 Report

I have no further comments.,